# Changes in Oral Health Behavior According to Dental Calculus Removal Health Insurance in Korea

**DOI:** 10.3390/bs13040315

**Published:** 2023-04-06

**Authors:** Yu-Rin Kim, Hyun-Kyung Kang

**Affiliations:** Department of Dental Hygiene, Silla University, 140 Baegyang-daero 700 Beon-gil, Sasang-gu, Busan 46958, Republic of Korea

**Keywords:** biofilms, national health insurance, oral health, periodontal disease

## Abstract

Background: Periodontal disease is a chronic disease that is increasing year by year. Korea also recognizes the seriousness of periodontal disease and has been applying preventive scaling in the National Health Insurance since 2013 to prevent it. Studies confirming the effectiveness of such insurance coverage are very scarce. Therefore, this study intended to confirm the effect of such policy by comparing and analyzing the oral health characteristics and oral health behavior of the South Koreans before and after the scaling insurance. Methods: For all the analyses, complex sampling analysis with the stratification variable, clustering variable, and weight was applied. For a total of 40,945 people, the demographic characteristics, oral health characteristics, dental clinic use, brushing, and use of oral care supplies were compared through chi-square tests. Results: Scaling insurance showed a positive effect (*p* < 0.05) on the unemployed and elderly people, who had lost their previous economic status; on smoking and willingness to quit smoking as well as on consultation on drinking problems; on dental-clinic use and oral examination in terms of utilization of dental clinics; and on brushing after lunch, before breakfast, and before sleeping. Conclusions: The study results showed that the scaling rate was universalized, and there was a positive effect on willingness to quit smoking and undergo oral examination. An active reimbursement policy for oral health education is needed, however, if a substantial change in oral health behavior is to be achieved.

## 1. Introduction

Periodontal disease is associated with hormonal and systemic diseases, as well as with age and gender, and is recognized as a serious disease, similar to other systemic diseases [1]. The World Health Organization (WHO) has set the goal of reducing the serious periodontal disease rate among adults in the oral area of Healthy People 2020, emphasizing the need for the management and treatment of periodontal disease [2]. In South Korea, the ratio of people with paradental cyst is 29.8% among the middle-aged and over 40% among the elderly, showing that periodontal disease is serious [3]. Periodontal disease is a chronic inflammatory disease caused by the host’s reaction to the biofilm formed on the surface of the teeth, which becomes hard dental calculus after some time, destroys the alveolar bone, and eventually causes tooth loss [4]. Therefore, to prevent the progression of periodontal disease, it is necessary to manage biofilms and dental calculus [5].

Scaling is a preventive treatment by an expert for removing supragingival and subgingival calculus, biofilms, and extrinsic pigmentation [6]. From July 2013, the health insurance began to apply once a year to adults over the age of 20, and from July 2017, the application was expanded to those over the age of 19 [7]. This allowed people to regularly have oral examinations, to detect and treat periodontal disease early, and to expect time and cost savings due to preventive treatment [8]. For this reason, Lee et al. [9] reported that “twice a year” was the most appropriate number of dental calculus removal insurance applications per year. As such, because dental calculus removal is covered by insurance, it has become a very important part of periodontal health for people. In general, after scaling is completed, professional oral health education is provided. The most important aspect of oral health education is education on brushing and oral care supplies according to the patient’s oral condition [10]. Such education is important for maintaining and promoting oral health after scaling, allowing people to do oral care by themselves. 

The use of oral care supplies can effectively manage biofilms that are not removed only by brushing. Therefore, it is important to educate people about selecting and using oral care supplies that best suit their oral environment [11]. In the results of a survey on oral care supplies for adults, however, the awareness rate was 58.3%, but the practice rate was only 31.3% [12]. Thus, the necessity is recognized, but the practice of using oral care supplies is low. As the national health insurance can be applied to preventive scaling once a year [13], it is necessary to examine the differences in the oral health characteristics, dental-clinic use, brushing, and use of oral care supplies of the South Korean people. The previous domestic studies on the national health insurance coverage of scaling include studies on the recognition of industry accident injury patients [14], the difference in the recognition between lay people and dental hygienists [15], the factors influencing the oral health beliefs on scaling performance by national health insurance coverage in the consumers [16], and analysis of the changes that transpired after scaling came to be covered by the national health insurance [17]. There is a dearth of research, however, on oral health characteristics and oral health behavior change based on the analysis of the cost difference. Therefore, the hypotheses of this study are as follows.

**H_0_.** 
*There will be no difference in oral health characteristics and oral health behaviors before and after dental calculus removal health insurance.*


**H_1._** 
*There will be difference in oral health characteristics and oral health behaviors before and after dental calculus removal health insurance.*


Accordingly, this study aimed to investigate the effect by comparing the differences in oral health characteristics and oral health behaviors before and after the once-a-year scaling insurance using data from the 5th–7th Korea National Health and Nutrition Examination Survey (KNHANES).

## 2. Materials and Methods

### 2.1. Study Design

This study used the 5th-round data of Korea National Health and Nutrition Examination Survey (KNHANES) conducted every year by the Centers for Disease Control and Prevention as data before the scaling insurance. Excluding the year 2013, when the insurance was applied, the 2nd and 3rd datasets of the 6th round and the 1st dataset of the 7th round were used as data after the scaling insurance (Figure 1). 

### 2.2. Setting

Data were collected from health insurance subscribers in Korea and analyzed, except for data from 2013, when tartar removal was covered by health insurance. Therefore, the year before tartar removal is covered by health insurance is from 2010 to 2012, and the year after tartar removal is covered by health insurance is from 2014 to 2016. KNHANES is a survey to produce government-designated statistics (Approval No. 117002) based on Article 17 of the Statistics Act. For the 5th round, the 1st year was 2010-02CON-21-C, the 2nd year was 2011-02CON-06-C, and the 3rd year was 2012-01EXP-01-2C. For the 6th round, the 2nd year was 2013-12EXP-03-5C. After the 3rd year, it fell under the study that is directly conducted by the government for the public welfare.

### 2.3. Study Participants

The participants were all persons subscribed to the National Health Insurance Corporation, and those who did not respond to general health examinations and oral health questionnaires were excluded. In addition, those who marked “don’t know” and “not applicable” in the responses were excluded. A total of 40,945 people were selected for the study: 7542 in 2010, 7252 in 2011, 6801 in 2012, 6420 in 2014, 6227 in 2015, and 6703 in 2016.

### 2.4. Variables

#### 2.4.1. Demographic Characteristics

Through the health survey of KNHANES, the gender, age, marriage, economic status, and income were examined. Age was classified into “<20”, “≥20 and <40”, “≥40 and <60”, and “≥60”. The income was classified into quintile, including “lower”, “lower-middle”, “middle”, “high-middle”, and “high”.

#### 2.4.2. Oral Health Characteristics

The oral health characteristics were examined through oral examination, recent consultation on drinking problems, and current smoking status. The willingness to quit smoking was measured based on a 4-point scale, and through reverse coding, the higher the score was, the greater the willingness to quit smoking. The chewing and speaking problems were measured based on a 5-point scale, and through reverse coding, the higher the score was, the more serious the problem.

The dental-clinic use was classified into “non-use” and “use” by dividing the use status into oral examination, periodontal treatment, simple caries treatment, dental pulp treatment, oral prevention, oral surgical treatment, trauma treatment, and prosthetic treatment.

#### 2.4.3. Oral Health Behavior

For brushing, whether brushing was performed was checked at 9 time points: yesterday, before and after breakfast, before and after lunch, before and after dinner, after snacking, and before sleeping. Oral care supplies were classified into dental floss, interdental toothbrush, toothpaste solution, electric toothbrush, and others to check their use status.

### 2.5. Data Sources/Measurement

The study compares oral health characteristics and oral health behaviors before and after calculus removal insurance coverage. As oral health characteristics, oral examination, use of dental clinic, recent drinking problem counseling, and current smoking status were confirmed. Furthermore, for chewing and speaking problems, the higher the score, the more serious the problem was. Oral health behaviors confirmed the timing of brushing teeth and the use of oral care products.

### 2.6. Bias

This study only confirmed differences in oral-related factors before and after health insurance coverage for tartar removal. Therefore, there is no potential source of bias as this is not a study to identify an influencing factor.

### 2.7. Study Size

This study was analyzed using the National Health and Nutrition Survey data provided by the Korea Centers for Disease Control and Prevention. Therefore, the subjects included Korean health insurance subscribers, and only those enrolled in 2013, when tartar removal was covered by insurance, were excluded from the subjects from 2010 to 2016.

### 2.8. Quantitative Variables

In the oral health characteristics, the intention to quit smoking was measured on a 4-point scale, and through reverse coding, higher scores meant higher intention to quit smoking. In addition, writing and speaking problems were measured on a 5-point scale, and the higher the score, the more serious the problem was evaluated through reverse coding.

### 2.9. Statistical Methods

The data were analyzed using IBM SPSS ver. 21.0 (IBM Co., Armonk, NY, USA). For all the analyses, complex sampling analysis was used with the stratification variable, clustering variable. For a total of 40,945 people (21,595 and 19,350 before and after the scaling insurance, respectively), the demographic characteristics, oral health characteristics, dental-clinic use, brushing, and use of oral care supplies were compared through chi-square tests, and the significance level of the statistical test was 0.05.

## 3. Results

### 3.1. Comparison of the Demographic Characteristics before and after the Once-a-Year Scaling Insurance

The proportions of both the males and the females were higher after the insurance application than before it. In terms of age, the proportion of those 39 years or younger was higher before the insurance application while the proportion of those 40 years or older was higher after. The proportion of those who were married was higher after the insurance application, but the proportion of those who were single was higher before. As for the economic activity, the proportion of those who were employed was higher before the insurance application while the proportion of the unemployed was higher after. In terms of income, the proportions of those who had “lower” and “lower-middle” incomes were higher before the insurance application while the proportions of those who had “middle”, “high-middle”, and “high” incomes were higher after it (Table 1). 

### 3.2. Comparison of the Oral Health Characteristics before and after the Once-a-Year Scaling Insurance Application

In terms of oral examination, the proportion of “yes” answers was higher after the insurance application while the proportion of “no” answers was higher before. As for the consultation on drinking problems for the last 1 year, the proportion of “yes” answers was higher before the insurance application while the proportion of “no” answers was higher after. As for current smoking, the proportion of “every day” answers was higher before the insurance application, and the proportions of the “sometimes” and “non-smoking” answers were higher after. In addition, the will to quit smoking was higher after insurance than before insurance. In terms of oral problems, the proportions of those with chewing and speaking problems were smaller after the insurance application than before it (Table 2).

### 3.3. Comparison of Dental-Clinic Use before and after the Once-a-Year Scaling Insurance Application

In terms of dental-clinic use, the proportion of “use” was higher after the insurance, and the proportion of “non-use” was higher before the insurance. The proportion of those with no oral examination experience was higher before the insurance application, and the proportion of those with oral-examination experience was higher after. The proportions of those who have had periodontal treatment, simple caries treatment, dental pulp treatment, oral prevention, oral surgical treatment, trauma treatment, and prosthetic treatment were all higher after the insurance application than before it (Table 3).

### 3.4. Comparison of Brushing before and after the Once-a-Year Scaling Insurance 

In terms of brushing yesterday, the proportions of “yes” and “no” answers were both higher after the insurance application, and the proportion of those not brushing after breakfast was higher after the insurance application while the proportion of those brushing after breakfast was higher before. The proportion of those not brushing after lunch was higher before the insurance application while the proportion of those brushing after lunch was higher. The proportion of those not brushing after dinner was higher after the insurance application while the proportion of those brushing after dinner was higher before. The proportion of those not brushing before breakfast was higher before the insurance application while the proportion of those brushing before breakfast was higher after. The proportions of those brushing and not brushing before lunch and before dinner were both higher after the insurance application. In addition, the proportions of those brushing and not brushing after snacking were both higher after the insurance application. The proportion of those not brushing before sleeping was higher before the insurance while the proportion of those brushing before sleeping was higher after the insurance application (Table 4).

### 3.5. Comparison of the Use of Oral Care Supplies before and after the Once-a-Year Scaling Insurance Application

The proportions of the “use” and “non-use” of dental floss, interdental toothbrush, toothpaste solution, electric toothbrush, and others were all higher after the insurance application than before it. There were significant differences in all the items except for interdental toothbrush and others (Table 5).

## 4. Discussion

In the early stage of periodontal disease, there is no discomfort; as such, people leave the disease untreated. If it is treated only after discomfort is felt, it takes considerable time and money to restore and rehabilitate the teeth [18]. Therefore, the early treatment and prevention of periodontal disease is most important. South Korea recognized the seriousness of periodontal disease and expanded the coverage of the national health insurance to include scaling once a year for all the citizens aged 19 years or older. The purpose of this study was to investigate the effect of such policy by comparing and analyzing the oral health characteristics and behaviors of the South Korean people before and after the once-a-year scaling insurance implemented in 2013. Of the total of 40,945 survey respondents, the proportions of males and females were both higher after the insurance application than before it, which is different from the past outcome of a higher proportion of females before the scaling insurance application. Thus, the effect of the scaling insurance application is considered leveled against gender. Among the age groups, the proportion of those 39 years old or younger was higher before the insurance application, and the proportion of those who were 40 years or older was higher after, showing a significant effect of drawing the dental-clinic use of elderly people with serious periodontal disease. The proportion of those who were single was higher before the insurance application while the proportion of those who were married was higher after the insurance application. As for income, the proportions of those with “lower” and “lower-middle” incomes were higher before the insurance application while the proportions of those with “middle”, “high-middle”, and “high” incomes were higher after, showing a consistent study outcome [19] that the healthcare-seeking behavior increases as the income level increases. The increased proportion of unemployed people after the insurance application, however, is considered a positive effect that lowers the threshold of dental-clinic use for the people who are not engaging in economic activities.

In terms of oral examination, the proportion of “yes” answers increased after the insurance application, and the proportions of chewing and speaking problems among the oral problems were smaller after the insurance application than before it. The once-a-year scaling increased the oral examination rate and had a positive effect on oral problems. When asked whether they had consulted for drinking problems, the proportion of “yes” answers was higher before the insurance application, but the proportion of “no” answers was higher after. It is believed that the consultation rate for drinking problems became lower because the people perceived the relationship between drinking and oral health. As for current smoking, the proportion of those who were smoking “every day” was higher before the insurance application, the proportions of those who were smoking “sometimes” and who were “non-smoking” were higher after, and the proportion of those who indicated a willingness to quit smoking was higher after. Tobacco contains about 4000 chemical components the exposure to which leads to arteriosclerosis. These include nicotine, which represents the toxicity of tobacco; tar, a major ingredient that causes lung cancer; and carbon monoxide, which causes oxygen deficiency [20]. These ingredients are known to cause extrinsic pigmentation in the teeth, and to be associated with bad breath, tartar formation, oral dryness, delayed wound healing, periodontitis, and tooth loss [21]. Therefore, it is thought that as the treatment for addictive smoking is performed in the dental clinic, and the education about smoking cessation can be provided by an expert, smoking cessation education is included in the oral health education that is given after once-a-year scaling, thus showing its effect. In particular, as smokers have more plaque and calculus than non-smokers [22], education about smoking cessation should be systematically implemented.

After the insurance application, the proportions of dental-clinic use and oral-examination experience were higher. This is because the one-a-year scaling makes people receive a regular oral examination at least once a year with preventive treatment for periodontal disease, so they may pay more attention to their oral condition and develop a higher need for oral health promotion. In terms of the reason for hospital use, there were no differences in the treatments because the proportions of those with experience and nonexperience in periodontal treatment, simple caries treatment, dental pulp treatment, oral prevention, oral surgical treatment, trauma treatment, and prosthetic treatment both increased after the insurance application.

Scaling is oral care provided by a professional, and self-oral care is important for sustaining oral health. A typical self-oral care regimen is brushing; however, a one-time brushing lesson is not sufficient, and it is more effective to teach it repeatedly at least five times, but the brushing lesson after scaling at the dental clinic is actually all that people usually have time for. In this study, the once-a-year scaling insurance application showed positive effects only after lunch, before breakfast, and before sleeping. There was no significant difference after breakfast, before and after dinner, before lunch, and after snacking. According to a study by Jo et al. [23], the ability to manage biofilms through brushing lessons increased gradually in their study over a period of five sessions when performed repeatedly at regular intervals, and it was even maintained overmore than 10 hospital visits thereafter. Based on this, it is appropriate to change the maximum number of claims for brushing lessons to five times or more, and if possible, it is necessary to adjust the insurance fee by including the brushing lesson to the once-a-year scaling insurance.

To manage oral health, it is essential to use oral care supplies according to a person’s own oral environment. Kim et al. [24] stated that it is necessary to provide information and educate people about oral care supplies. Thus, it is necessary to provide education on dental floss and interdental toothbrushes that clean the interdental space where a paradental cyst is formed first and then deepens, as well as education on the water flosser, which can prevent the formation of biofilms by blowing water between the teeth [25]. In this study, there was no positive difference in the use of the oral care supplies included in the oral health education performed after once-a-year scaling, including dental floss, interdental toothbrush, toothpaste solution, electric toothbrush, and others. Thus, dental professionals need to recognize the importance of oral health education, and an institutional policy on oral health education is needed.

Although once-a-year scaling insurance had positive effects on the willingness to quit smoking, oral examination, and universalization of the scaling rate, the oral health education and preventive-treatment items for maintaining and promoting oral health are unsatisfactory. The national health insurance needs to cover the toothpick method [26], which is the typical brushing method for interdental surface periodontal management. It will be necessary to continually expand the policy of applying dental preventive care benefits at a level that encompasses preventive measures such as oral hygiene management, oral health education, and periodontal disease management, similar to the system of periodontal maintenance in Japan, rather than limiting the number of annual benefits and the application of a single preventive benefit. As this study analyzed only the results of the 5th–7th Korea National Health and Nutrition Examination Survey (KNHANES), it has a limitation in generalizing the study results. In addition, since the National Health and Nutrition Examination Survey is in the form of a questionnaire, there is a high possibility for one’s own subjective involvement, and it is difficult to confirm a clear causal relationship due to the characteristics of a cross-sectional survey. Therefore, in future research, it will be necessary to integrate the oral examination results of the subjects and the health survey results of the patients, and to expand the connection with the characteristics, so that in-depth analysis can be made.

## 5. Conclusions

This study will contribute to understanding the effect of insurance policy as a study that confirms the change in oral health characteristics and behavior before and after dental calculus removal is applied to insurance. First, with the application of dental calculus removal insurance, the number of people receiving oral examinations and preventive treatment will increase, enabling early detection and prevention of periodontal disease. Second, as the frequency of brushing increases due to the effect of dental calculus removal insurance, oral health education conducted during dental calculus removal will be important. Third, it is thought that oral self-care will be possible as the use of oral aid products increases with the application of dental calculus removal insurance. An active reimbursement policy for oral health education is needed, however, if a substantial change in oral health behavior is to occur.

## Figures and Tables

**Figure 1 behavsci-13-00315-f001:**
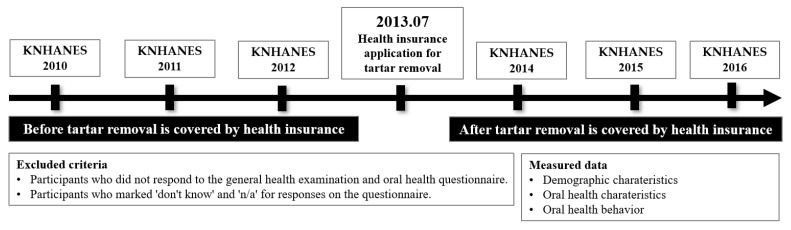
Study design.

**Table 1 behavsci-13-00315-t001:** Comparison of the subjects’ demographic characteristics (Unit: %).

		Before	After	*p* *
Gender	Male	9330 (49.4)	8400 (50.6)	0.871
	Female	1226 (49.3)	10,950 (50.7)	
	Total	21,595 (49.4)	19,350 (50.6)	
Age	<20	5365 (52.3)	4322 (47.7)	0.000
	20–39	4506 (50.7)	3939 (49.3)	
	40–59	5942 (48.2)	5518 (51.8)	
	60≤	5782 (45.8)	5571 (54.2)	
	Total	21,595 (49.4)	19,350 (50.6)	
Marital status	Married	14,322 (48.7)	13,003 (51.3)	0.012
	Single	7223 (50.3)	6346 (49.7)	
	Total	21,545 (49.3)	19,349 (50.7)	
Economic activity	Active	9151 (50.5)	8120 (49.5)	0.148
	None	7519 (49.5)	6591 (50.5)	
	Total	16,670 (50.1)	14,711 (49.9)	
Household income	Lower	4199 (51.8)	3822 (48.2)	0.000
	Lower-middle	4339 (51.0)	3845 (49.0)	
	Middle	4298 (49.3)	3840 (50.7)	
	High-middle	4259 (47.5)	3891 (52.5)	
	High	4239 (45.8)	3864 (54.2)	
	Total	21,334 (49.2)	19,262 (50.8)	

* Using complex-samples chi-square test.

**Table 2 behavsci-13-00315-t002:** Comparison of the subjects’ oral-related characteristics (Unit: %).

		Before	After	*p* *
Oral examination	No	14,599 (52.3)	11,761 (47.7)	0.000
	Yes	6338 (43.5)	6895 (56.5)	
	Total	20,937 (49.5)	18,656 (50.5)	
Consultation on drinking problems	No	13,350 (47.4)	13,227 (52.6)	0.000
	Yes	181 (67.0)	89 (33.0)	
	Total	13,531 (47.6)	13,316 (52.4)	
Currently smoking	Every day	2623 (53.4)	2126 (46.6)	0.000
	Sometimes	300 (45.0)	347 (55.0)	
	Non-smoking	3242 (48.3)	3000 (51.7)	
	Total	6165 (50.0)	5475 (49.5)	
Willingness to quit smoking **		2.28 ± 0.02	2.37 ± 0.03	0.008
Chewing problem **		2.31 ± 0.02	2.22 ± 0.01	0.000
Speaking problem **		1.64 ± 0.02	1.62 ± 0.001	0.110

* Using complex-samples chi-square test, ** using independent t-test with complex samples.

**Table 3 behavsci-13-00315-t003:** Comparison of the subjects’ dental-clinic use (Unit: %).

		Before	After	*p* *
Dental-clinic use	Used	4305 (27.2)	10,304 (72.8)	0.000
	Unused	16,633 (62.8)	8352 (37.2)	
	Total	20,938 (49.4)	18,656 (50.6)	
Oral examination	Inexperienced	2365 (50.7)	1958 (49.3)	0.000
	Experienced	1990 (17.8)	8346 (82.2)	
	Total	4355 (27.4)	10,304 (72.6)	
Periodontal treatment	Inexperienced	3787 (28.6)	8345 (71.4)	0.000
	Experienced	568 (21.9)	1959 (78.1)	
	Total	4355 (27.4)	10,304 (72.6)	
Caries treatment	Inexperienced	3127 (28.1)	7068 (71.9)	0.022
	Experienced	1228 (25.9)	3236 (74.1)	
	Total	4355 (27.4)	10,304 (72.6)	
Dental pulp treatment	Inexperienced	3764 (28.4)	8369 (71.6)	0.000
	Experienced	591 (22.9)	1935 (77.1)	
	Total	4355 (27.4)	10,304 (72.6)	
Oral prevention	Inexperienced	3488 (35.2)	5889 (64.8)	0.000
	Experienced	867 (14.9)	4415 (85.1)	
	Total	4355 (27.4)	10,304 (72.6)	
Oral surgical treatment	Inexperienced	3909 (27.9)	9028 (72.1)	0.012
	Experienced	446 (24.2)	1276 (75.8)	
	Total	4355 (27.4)	10,304 (72.6)	
Trauma treatment	Inexperienced	4260 (27.4)	10,089 (72.6)	0.223
	Experienced	95 (31.0)	215 (69.0)	
	Total	4355 (27.4)	10,304 (72.6)	
Prosthetic treatment	Inexperienced	3494 (28.0)	8169 (72.0)	0.006
	Experienced	861 (24.8)	2135 (75.2)	
	Total	4355 (27.4)	10,304 (72.6)	

* Using complex-samples chi-square test.

**Table 4 behavsci-13-00315-t004:** Comparison of the subjects’ brushing (Unit: %).

		Before	After	*p* *
Yesterday brushing	No	364 (43.3)	410 (56.7)	0.007
	Yes	20,573 (49.5)	18,251 (50.5)	
	Total	20,937 (49.4)	18,661 (50.6)	
Before breakfast	No	16,354 (54.1)	12,241 (45.9)	0.000
	Yes	4270 (38.9)	6010 (61.1)	
	Total	20,624 (49.6)	18,251 (50.4)	
After breakfast	No	5457 (43.7)	6175 (56.3)	0.000
	Yes	15,167 (52.5)	12,076 (47.5)	
	Total	20,624 (49.6)	18,251 (50.4)	
Before lunch	No	20,407 (49.7)	17,987 (50.3)	0.014
	Yes	217 (41.1)	264 (58.9)	
	Total	20,624 (49.6)	18,251 (50.4)	
After lunch	No	12,231 (53.3)	9515 (46.7)	0.000
	Yes	8393 (45.1)	8736 (54.9)	
	Total	20,624 (49.6)	18,251 (50.4)	
Before dinner	No	20,193 (49.7)	17,819 (50.3)	0.418
	Yes	431 (47.8)	432 (52.2)	
	Total	20,624 (49.6)	18,251 (50.4)	
After dinner	No	7183 (43.1)	8204 (56.9)	0.000
	Yes	13,441 (53.9)	10,047 (46.1)	
	Total	20,624 (49.6)	18,251 (50.4)	
After snacking	No	20,182 (49.9)	17,651 (50.1)	0.000
	Yes	442 (40.1)	600 (59.9)	
	Total	20,624 (49.6)	18,251 (50.4)	
Before sleeping	No	14,062 (57.1)	9478 (42.9)	0.000
	Yes	6562 (39.3)	8773 (60.7)	
	Total	20,624 (49.6)	18,251 (50.4)	

* Using complex-samples chi-square test.

**Table 5 behavsci-13-00315-t005:** Comparison of the subjects’ oral care supplies (Unit: %).

		Before	After	*p* *
Dental floss	Unused	7367 (33.5)	12,619 (66.5)	0.000
	Use	2383 (37.8)	3326 (62.2)	
Interdental toothbrush	Unused	7583 (34.2)	13,054 (65.8)	0.129
	Use	2167 (35.6)	2893 (64.4)	
Toothpaste solution	Unused	8223 (35.3)	12,592 (64.7)	0.000
	Use	1527 (31.2)	3355 (68.8)	
Electric toothbrush	Unused	9040 (33.8)	15,215 (66.2)	0.000
	Use	710 (45.5)	732 (54.5)	
Other	Unused	9073 (34.4)	15,004 (65.6)	0.538
	Use	677 (35.4)	943 (64.6)	
	Total	9750 (34.5)	15,947 (65.5)	

* Using complex-samples chi-square test.

## Data Availability

The data presented in this study are available on request from the corresponding author.

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
