# Peer review of "Changes in Oral Health Behavior According to Dental Calculus Removal Health Insurance in Korea"

_behavsci, 2023, doi:10.3390/bs13040315_

Round 1

Reviewer 1 Report

I think your research will have a positive impact on Korea's health insurance system.  It is very interesting that tartar removal is covered by insurance once a year as a system for oral health of Koreans. I think your study is very meaningful as a study to confirm the effectiveness of the system.

However, if you modify it according to the following contents, it will be a better result of the study.

1. In the discussion, present the limitations of your study in detail.

2. Please add about future research directions.

3. Get English proofreading by a native speaker.

Author Response

Reviewer 1

I think your research will have a positive impact on Korea's health insurance system.  It is very interesting that tartar removal is covered by insurance once a year as a system for oral health of Koreans. I think your study is very meaningful as a study to confirm the effectiveness of the system.

However, if you modify it according to the following contents, it will be a better result of the study.

  1. In the discussion, present the limitations of your study in detail.

→ We added limitations in the Discussion. [In addition, since the National Health and Nutrition Examination Survey is in the form of a questionnaire, there is a high possibility that one's own subjective involvement may be involved, and it is difficult to confirm a clear causal relationship due to the characteristics of a cross-sectional survey.]

  1. Please add about future research directions.

→ At the end of the discussion, the future research direction is written. [in the future research, it is necessary to integrate the oral examination results of the subjects and the health survey results of the patients, and to expand the connection with the characteristics, so that in-depth analysis can be made.]

  1. Get English proofreading by a native speaker.

→ In your opinion, we have English proofread by native speakers. thank you.

Reviewer 2 Report

Dear Authors,

Thank you for submitting this manuscript. I think the paper is quite interesting because it refers to a very important topic: changes in oral health behavior, according to dental calculus removal health insurance. I would like to suggest some points to the Authors:

1. The abstract should include a short statement on the current research gap and question to show why this study is unique and worthy of publication.

2. In the Abstract section, please follow the instructions for authors – background, introduction, materials and methods, results, discussion, and conclusion.

3. Line 8 – there is a repetition, please revise this sentence.

4. Line 12 – there is a repetition, please revise this sentence.

5. In the Introduction, line 30: please add more references.

6. Lines 46 – 48: This sentence is too long, please revise it.

7. Line 53 – Please add more references.

8. The authors should add the null and working hypotheses and highlight them by adding "H0" and "H1"

9. Line 130 – 133: This sentence is too long, please revise it.

10. I would suggest presenting the results by column charts instead of tables.

11. The authors should summarize the significant findings in bullets for clarity in the Conclusion section.

Thank you in advance for all the corrections.

Best regards!

Author Response

Reviewer 2

Dear Authors,

Thank you for submitting this manuscript. I think the paper is quite interesting because it refers to a very important topic: changes in oral health behavior, according to dental calculus removal health insurance. I would like to suggest some points to the Authors:

  1. The abstract should include a short statement on the current research gap and question to show why this study is unique and worthy of publication.

→ Thanks for your views. We have edited the abstract.

  1. In the Abstract section, please follow the instructions for authors – background, introduction, materials and methods, results, discussion, and conclusion.

→ Thanks for your views. We have edited the abstract.

  1. Line 8 – there is a repetition, please revise this sentence.

→ We have corrected the content.

  1. Line 12 – there is a repetition, please revise this sentence.

→ We have corrected the content.

  1. In the Introduction, line 30: please add more references.

→ We have added a reference.

  1. Lines 46 – 48: This sentence is too long, please revise it.

→ We modified it into two sentences to help the reader's understanding.

  1. Line 53 – Please add more references.

→ We have added a reference.

  1. The authors should add the null and working hypotheses and highlight them by adding "H0" and "H1"

→ We added the null and alternative hypotheses according to your opinion. [Therefore, the hypotheses of this study are as follows.

H0: There will be no difference in oral health characteristics and oral health behaviors before and after dental calculus removal health insurance.

H1: There will be difference in oral health characteristics and oral health behaviors before and after dental calculus removal health insurance.]

  1. Line 130 – 133: This sentence is too long, please revise it.

→ We modified it into two sentences to help the reader's understanding. Also, we corrected the text.

  1. I would suggest presenting the results by column charts instead of tables.

→ We respect your views. However, we believe that a table is more appropriate than a bar graph to record significance (p-value). Thank you for your consideration.

  1. The authors should summarize the significant findings in bullets for clarity in the Conclusion section.

→ Thanks for your views. We have added a summary of key findings.

Thank you in advance for all the corrections.

Best regards!

→ Thanks for your views. We will try to do better research.

Reviewer 3 Report

The manuscript titled "Changes in Oral Health Behaviour according to Dental Calculus Removal Health Insurance in Korea" is a research article focusing on the effect of such policy by comparing and analysing the oral health characteristics and oral health behaviour of the South Koreans before and after the scaling insurance. The manuscript is well structured and scientifically sound. However, there are few suggestions needs to be addressed. The manuscript needs a minor revision and the following are the comments needed to be addressed. 

Minor Comments:

1.     The introduction needs to be modified. The authors haven’t mentioned clearly the statement of the problem like, why the suggested study is better choice to replace the existing studies. Try to add a few works of literature to support the statement. 

2.     The background of dental insurance need to be still elaborated for a better understanding of the importance of the insurance and oral health.

3.     Why gender group sample size has difference? Does it change the results?

4.     Was the time point intervals followed for hoe many days?

5.     Whether night time brushing was considered?

6.     What are the justification for using these time points and the time interval of the study? Is there any references?

Author Response

Reviewer 3

The manuscript titled "Changes in Oral Health Behaviour according to Dental Calculus Removal Health Insurance in Korea" is a research article focusing on the effect of such policy by comparing and analysing the oral health characteristics and oral health behaviour of the South Koreans before and after the scaling insurance. The manuscript is well structured and scientifically sound. However, there are few suggestions needs to be addressed. The manuscript needs a minor revision and the following are the comments needed to be addressed. 

Minor Comments:

  1. The introduction needs to be modified. The authors haven’t mentioned clearly the statement of the problem like, why the suggested study is better choice to replace the existing studies. Try to add a few works of literature to support the statement. 

→ In line with your opinion, we have added information about tartar removal insurance coverage in the introduction.

  1. The background of dental insurance need to be still elaborated for a better understanding of the importance of the insurance and oral health.

→ We have added details of the process of starting and expanding dental calculus removal health insurance coverage. thank you.

  1. Why gender group sample size has difference? Does it change the results?

→ Our study used data from the National Health and Nutrition Examination Survey. The sample size may vary according to each survey participant. In addition, since the composite sample analysis was performed by applying weights, there may be differences in the ratio according to each group. Thank you for your consideration.

  1. Was the time point intervals followed for hoe many days?

→ Our study was a cross-sectional study, confirming each time point. Therefore, only differences between the two groups, not causality, was confirmed.

  1. Whether night time brushing was considered?

→ We made sure to brush our teeth before going to bed. This is presented in Table 4.

  1. What are the justification for using these time points and the time interval of the study? Is there any references?

→ Dental calculus removal was covered by insurance in July 2013. Therefore, we believe that not including 2013 is appropriate for the pre-insured and post-insured groups. There are no references related to this, and the analysis was conducted in consideration of the method to minimize bias during the research method. Thank you for your consideration.